ecology/environmental science/microbiology

marine mammal, microorganism, phylogeny, SSU ribosomal RNA gene, bacteria

**Author for correspondence:**
Amy Apprill
e-mail: aapprill@whoi.edu

# Marine mammal skin microbiotas are influenced by host phylogeny

Amy Apprill[1], Carolyn A. Miller[1], Amy M. Van Cise[2], Jana M. U'Ren[3], Matthew S. Leslie[4], Laura Weber[1], Robin W. Baird[5], Jooke Robbins[6], Scott Landry[6], Andrea Bogomolni[2] and Gordon Waring[7]

[1]Marine Chemistry and Geochemistry Department, and [2]Biology Department, Woods Hole Oceanographic Institution, Woods Hole, MA, USA
[3]BIO5 Institute and Department of Biosystems Engineering, University of Arizona, Tucson, AZ, USA
[4]Biology Department, Swarthmore College, Swarthmore, PA, USA
[5]Cascadia Research Collective, Olympia, WA, USA
[6]Center for Coastal Studies, Provincetown, MA, USA
[7]NOAA Northeast Fisheries Science Center, Protected Species Branch, Woods Hole, MA, USA

AA, 0000-0002-4249-2977; AMVC, 0000-0002-0613-4072;
MSL, 0000-0002-2536-6020; LW, 0000-0003-4957-2758;
RWB, 0000-0002-9419-6336; JR, 0000-0002-6382-722X

Skin-associated microorganisms have been shown to play a role in immune function and disease of humans, but are understudied in marine mammals, a diverse animal group that serve as sentinels of ocean health. We examined the microbiota associated with 75 epidermal samples opportunistically collected from nine species within four marine mammal families, including: Balaenopteridae (sei and fin whales), Phocidae (harbour seal), Physeteridae (sperm whales) and Delphinidae (bottlenose dolphins, pantropical spotted dolphins, rough-toothed dolphins, short-finned pilot whales and melon-headed whales). The skin was sampled from free-ranging animals in Hawai'i (Pacific Ocean) and off the east coast of the United States (Atlantic Ocean), and the composition of the bacterial community was examined using the sequencing of partial small subunit (SSU) ribosomal RNA genes. Skin microbiotas were significantly different among host species and taxonomic families, and microbial community distance was positively correlated with mitochondrial-based host genetic divergence. The oceanic location could play a role in skin microbiota variation, but skin from species sampled in both locations is necessary to determine this influence. These data suggest that a phylosymbiotic relationship may exist between microbiota and their marine mammal hosts,

potentially providing specific health and immune-related functions that contribute to the success of these animals in diverse ocean ecosystems.

## 1. Introduction

Marine mammals play essential roles in marine ecosystems as predators and primary and secondary consumers, and can be sentinels of ocean health [1]. There are 129 species of marine mammals spanning 16 families that comprise diverse taxa, such as dolphins, whales, seals, porpoises, manatees, walruses and polar bears, among others [2]. Many marine mammal species were hunted to very low numbers [3] and some populations, especially certain large baleen whales, have been slow to recover from extreme levels of exploitation. Today, marine mammals are susceptible to direct human impacts, including vessel strikes, sound pollution, fisheries bycatch, entanglement in fishing gear, and indirect impacts from climate and other ocean changes [1,4]. Accordingly, understanding the health of marine mammals and their response to disturbance is a central goal for the management and conservation of populations and the ecosystems to which they belong.

Microbial communities, or microbiotas, are composed of a diverse assemblage of cells, including bacteria, archaea, fungi and protists, which can play vital and active roles in maintaining normal functioning and health of humans and other animals [5,6]. Studies in a handful of marine mammal species show evidence of highly diverse gut, oral and respiratory bacterial assemblages that vary in concordance with host species, as well as diet and habitat [7–9]. Similar to connections between human gut microbiota and diet [10], the microbiotas associated with marine mammals may provide critical insights into animal health and ecology; however, the microbial communities of most marine mammals are unexplored [8].

Skin is the largest organ of marine mammals and it is their first level of defence against pathogens. Their skin is routinely in contact with the microorganisms in their immediate environment. In other mammals, skin-associated microbes play important roles in resisting the colonization of the skin by opportunistic invaders and refining the immune system [11–13]. Core epidermal microbiotas are consistent across isolated populations of humpback whales (*Megaptera novaengliae*), which may be indicative of a specific and refined functional relationship between hosts and their microbial communities [14,15]. In marine fish, as well as some orders of terrestrial mammals, phylosymbiotic or microbial–host evolutionary patterns [16] were recently found to structure skin microbial community composition [17,18], thus providing a mechanism driving microbial specificity in these and possibly other taxa.

Here, we present an examination of the skin microbiotas from nine marine mammal species residing within four families spanning the cetaceans (Physeteridae, Delphinidae and Balaenopteridae) and pinnipeds (Phocidae). The study was opportunistic, relying on existing samples from free-ranging animals in two oceans and sequencing protocols established by the Earth Microbiome Project (EMP). Our results suggest that host species identity and phylogenetic relationships both influence the composition of microorganisms associated with the skin of the species examined. Further, shared bacterial taxa were identified among marine mammal species, suggesting that commonalities in the skin may support the growth of bacterial cells across these diverse ocean mammals.

## 2. Material and methods

Samples of the epidermis (herein referred to as 'skin') were analysed from 75 free-ranging individuals spanning nine species from four families (Phocidae, Balaenopteridae, Delphinidae and Physeteridae) off the coast of Hawai'i and the eastern USA (table 1; electronic supplementary material, table S1). Samples from cetaceans were obtained from the flank of wild animals at sea using biopsy sampling techniques [19]. For the harbour seals (*Phoca vitulina*), skin biopsy samples were taken from the hind flipper when the animals were on shore (and subsequently tagged for a non-related study). Samples were placed on ice in the field and later frozen to at least −80°C. They were subsampled for this study using sterile tools. For each sample, approximately 25 mg of skin was used to extract microbial DNA using the DNeasy Blood and Tissue kit (Qiagen), as previously described [20], and DNA was quantified using the Qubit fluorescent assay (Invitrogen). Subsequent analyses including polymerase chain reaction (PCR), amplicon barcoding and sequencing were conducted by the EMP using an established methodology (http://www.earthmicrobiome.org/emp-standard-protocols/16s/) [21,22].

**Table 1.** Samples of marine mammals used to assess skin microbiotas were from 75 free-ranging animals representing four families and nine species.

| family (n) | species | common name (n) | location |
|---|---|---|---|
| Phocidae (21) | *Phoca vitulina* | harbour seal (21) | US east coast |
| Balaenopteridae (18) | *Balaenoptera physalus* | fin whale (15) | US east coast |
| | *Balaenoptera borealis* | sei whale (3) | US east coast |
| Delphinidae (30) | *Globicephala macrorhynchus* | short-finned pilot whale (15) | Hawai'i |
| | *Peponocephala electra* | melon-headed whale (2) | Hawai'i |
| | *Stenella attenuata* | pantropical spotted dolphin (6) | Hawai'i |
| | *Steno bredanensis* | rough-toothed dolphin (4) | Hawai'i |
| | *Tursiops truncatus* | bottlenose dolphin (3) | Hawai'i |
| Physeteridae (6) | *Physeter macrocephalus* | sperm whale (6) | Hawai'i |

In short, the hypervariable IV region of the small subunit ribosomal RNA (SSU rRNA) gene was PCR amplified using 515F (5′-GTGCCAGCMGCCGCGGTAA-3′) and 806R (5′-GGACTACHVGGGTW TCTAAT-3′) primers [21] with sample-specific barcodes (without enhanced coverage of the SAR11 clade or Thaumarchaeota [23,24]). Reactions (25 µl) consisted of: sterile PCR-grade water (13 µl), Platinum Hot Start PCR master mix (10 µl; ThermoFisher), 0.5 µl of each primer and 1 µl of template DNA. The reaction consisted of an initial denaturation at 94°C for 3 min followed by 35 cycles of 94°C for 45 s, 50°C for 60 s, 72°C for 90 s and concluded with 72°C for 10 min. Amplified samples were quantified using the Quanti-iT PicoGreen dsDNA Assay Kit (ThermoFisher/Invitrogen), pooled into equimolar ratios, cleaned using the UltraClean PCR Clean-Up Kit (MoBio Laboratories) and sequenced using the HiSeq platform (Illumina). Initial data processing was conducted by the EMP using standard protocols and included demultiplexing and quality filtering using the QIIME v. 1.5.0 [25], resulting in 19 455 833 sequences. Subsequent analysis was performed in mothur v. 1.33.3, including chimera detection and removal using UCHIME and detection and removal of Eukaryota, mitochondria and chloroplasts using the k-nearest neighbour algorithm with the SILVA SSU Ref database (v. 119) [26]. Minimum entropy decomposition (MED; v. 1.7) [27] was used to group reads into 365 MED nodes. Representative reads from the MEDs identified as unclassified Bacteria were further examined using NCBI's BLAST, and 71 MED nodes representing 1 386 502 sequences were identified as marine mammal mitochondria, and thus removed prior to the subsequent analysis. The final dataset of 75 samples all had more than 3000 sequences (electronic supplementary material, table S1). Hierarchical clustering, non-metric multidimensional scaling (nMDS) analysis and permutational multivariate analysis of variance (PERMANOVA) [28] statistical comparisons of the sequences were computed in Primer-E [29] using a distance matrix of Bray–Curtis dissimilarity and the group average clustering method for all samples, with Monte Carlo corrections applied. We conducted resemblance-based permutation tests (i.e. PERMDISP) to test the null hypothesis that average within-group dispersion is equivalent among groups [30], as differences in multivariate dispersion can affect PERMANOVA results [31]. Dispersion is measured as the average distance to the group centroid. PERMDISP and similarities and percentages routine (SIMPER) analyses were also concluded in Primer-E.

The sequences were not subsampled due to issues with this practice [32], and the following analysis was conducted to support this decision. PERMANOVA analysis of sequences grouped into categories by the number of sequences generated for each sample (i.e. I: 3000–9999 sequences; II: 10 000–59 999; III: 50 000–99 999; IV: 100 000+) were compared to a Bray–Curtis dissimilarity matrix of host-specific microbiotas in a pairwise (sequence category: sequence category) fashion for each marine mammal species. This analysis demonstrated that sequencing depth was not significantly related to microbiota structure ($p > 0.05$) in 16 of the 17 comparisons, with the only significant comparison of categories II and IV (10 000–59 999 compared to 100 000+ sequences) in the harbour seals ($t = 2.133$; $p = 0.008$).

To infer the phylogenetic relationships and genetic distances among host marine mammals, reference mitochondrial genomes from each species were obtained from NCBI (electronic supplementary material, table S2). Sequences were aligned in Geneious Prime (v. 2019.2.1, BioMatters), using the Clustal W alignment algorithm with default parameters [33]. Mitogenome tree topology was constructed using a

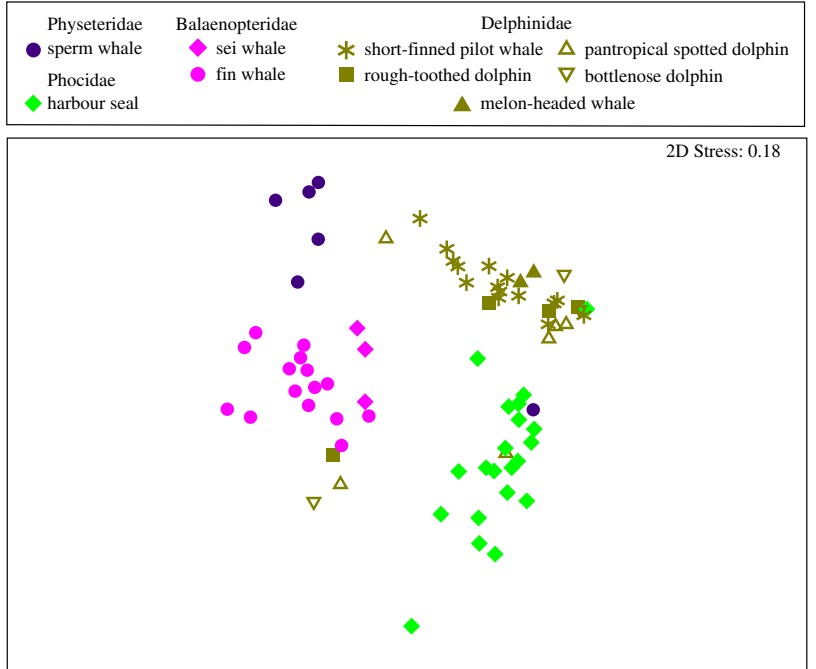

**Figure 1.** Marine mammal skin microbial communities are related to host family and species identity. Non-metric multidimensional scaling (nMDS) analysis of minimum entropy decomposition (MED) nodes compared using Bray–Curtis dissimilarity shows the similarity of microbial community composition among marine mammal species (denoted by shapes) residing in the same family (denoted by colour).

Bayesian maximum clade credibility (MCC) phylogenetic approach implemented in BEAST v. 1.10.4 [34]. The harbour seal was rooted as the outgroup, and monophyletic priors were set for all cetaceans, as well as for odontocetes and mysticetes. Model parameters included an uncorrelated relaxed clock to allow for variable substitution rates among the different species, a general time reversible (GTR) substitution model [35] with Gamma plus invariant site heterogeneity, and the Yule Process tree prior [36,37], appropriate for models with multiple species. Markov chain Monte Carlo (MCMC) was run with 10 million steps, sampled every 1000 steps. TreeAnnotator v. 1.10.4 [38] in the BEAST cluster was used to generate the MCC tree, after the removal of the first 10% of the MCMC samples, and the tree was visualized in FigTree v. 1.4.4 [39]. A Mantel test [40] was performed in mothur (using default parameters) to test for a correlation between host genetic distance (as estimated using mitochondrial genomes) and microbiota dissimilarity (using Bray–Curtis dissimilarity) using the Pearson correlation coefficient.

# 3. Results

Although the SSU rRNA gene primers targeted both Bacteria and Archaea, the sequence reads recovered were all from the domain Bacteria. Analysis of the composition of the bacterial SSU rRNA genes using nMDS comparison of Bray–Curtis dissimilarity MED node groupings demonstrated that the skin microbiotas clustered according to the host family level of taxonomy (figure 1). PERMANOVA analysis of these MED nodes verified that the marine mammal families possessed significantly distinct skin microbiotas ($p = 0.001$), and the PERMDISP analysis provided confidence that this result was not due to differences in dispersion among groups ($p = 0.794$; table 2). We observed significant differences in skin microbial communities between each marine mammal family ($p < 0.001$ for all PERMANOVA pairwise comparisons; table 3) and between host species (PERMANOVA, $p = 0.001$). The degree of β dispersion did not differ among species (PERMDISP $p = 0.463$; table 2). The majority of species in different families had significantly distinct skin microbiotas (i.e. 92% of comparisons), whereas only 18% of species within the same family had significantly distinct skin microbiota composition ($p < 0.05$ PERMANOVA with Monte Carlo corrections). For these comparisons, we failed to reject the null hypothesis of homogeneity of multivariate dispersion among groups (PERMDISP; see electronic supplementary material, table S3), thus providing confidence in the outcome of PERMANOVA comparisons.

**Table 2.** Statistical analysis of the effect of host taxonomy on the composition and degree of in-group β dispersion of the skin microbiotas (PERMANOVA *$p < 0.05$ as significantly distinct; PERMDISP $p > 0.05$). d.f., degrees of freedom; Sum sq, sum of squares; pseudo-*F*, *F*-value by permutation; *p*-values based on 999 permutations with Monte Carlo correction.

| test | samples | PERMANOVA | | | | | PERMDISP | |
|---|---|---|---|---|---|---|---|---|
| | | d.f. | Sum sq | pseudo-*F* | $R^2$ | *p*-value (Monte Carlo correction) | *F*-value | *p*-value |
| family level | four families (table 1) | 3 | 90 923 | 14.674 | 40.24 | 0.001* | 0.483 | 0.794 |
| species level (all samples) | nine species (table 1) | 8 | $1.10 \times 10^5$ | 7.12 | 39.17 | 0.001* | 1.672 | 0.463 |

**Table 3.** Pairwise statistical comparison of the effect of family-level host taxonomy on the composition and degree of in-group β dispersion of the skin microbiotas (pairwise PERMANOVA *$p < 0.05$ as significantly distinct; PERMDISP $p > 0.05$).

| comparison | PERMANOVA | | | | PERMDISP | |
|---|---|---|---|---|---|---|
| | *t* | *p* (perm) | unique perms | *p* (Monte Carlo correction) | *t* | *p*-value |
| Delphinidae, Balaenopteridae | 4.5797 | 0.001 | 999 | 0.001* | 1.2438 | 0.29 |
| Delphinidae, Phocidae | 3.9511 | 0.001 | 998 | 0.001* | 0.5153 | 0.656 |
| Delphinidae, Physeteridae | 2.797 | 0.001 | 999 | 0.001* | 0.5299 | 0.657 |
| Balaenopteridae, Phocidae | 4.5392 | 0.001 | 999 | 0.001* | 0.6916 | 0.533 |
| Balaenopteridae, Physeteridae | 3.0176 | 0.001 | 995 | 0.001* | 0.1879 | 0.903 |
| Phocidae, Physeteridae | 2.8924 | 0.002 | 997 | 0.001* | 0.2206 | 0.867 |

We observed a strong congruence between the host mitochondrial phylogeny and the similarity of the skin microbiotas (figure 2). A Mantel test identified a significant correlation between host genetic distance based on mitochondrial genomes and the composition of the skin microbiotas ($r = 0.77$, $p = 0.00001$). Similarity percentage analysis (SIMPER) [41] was used to examine specific MED nodes that contributed to family phylogeny-based differences, and this analysis indicated that the skin microbiotas of the baleen whales were shaped by distinct microbial lineages compared to the other marine mammals, including undescribed members of the Moraxellaceae and Cardiobacteriaceae families of Gammaproteobacteria (electronic supplementary material, table S4). Different *Psychrobacter* MEDs contributed to the microbial community structure of marine mammals affiliated with the Phocidae and Physeteridae families (electronic supplementary material, table S4). The same MED of *Pseudomonas* was well represented in the skin microbiotas of Delphinidae- and Phocidae-affiliated animals. Generally, the same MEDs contributing to the family-based microbial community structure also contributed to the species-based microbiotas (electronic supplementary material, table S5).

# 4. Discussion

This study provides a novel comparison of the skin microbiotas of the largest number of marine mammal species examined to date. We found that taxonomy (species and family) of the marine mammal hosts strongly influenced the composition of the skin microbiotas and that specific bacterial lineages contributed to this relationship. Further, the correlation of skin microbial composition with host genetic distance suggests the potential for phylosymbiosis [16], conserved evolutionary patterns among marine mammal hosts and the microorganisms residing on their skin.

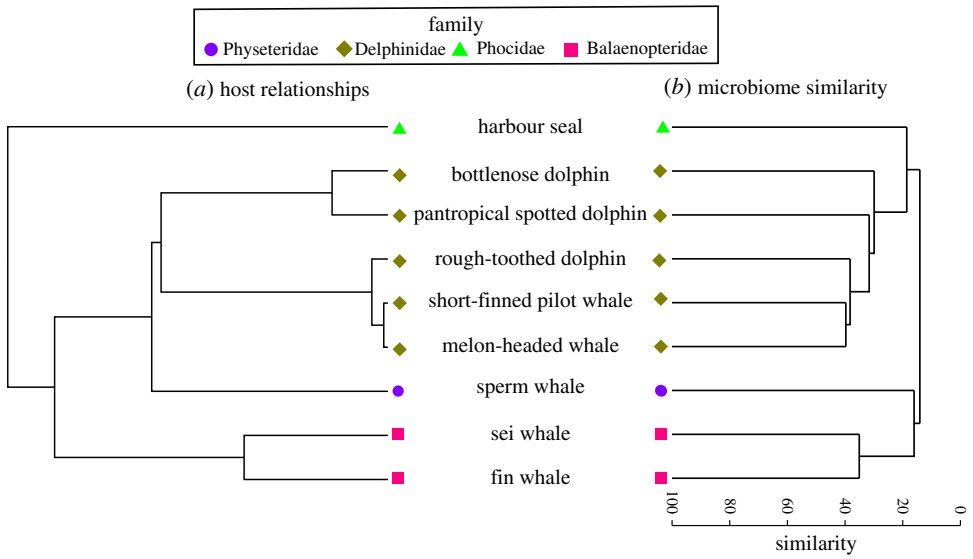

**Figure 2.** Evolutionary relationships among marine mammal hosts (*a*) are highly congruent with the clustering of skin microbial community similarity from free-ranging marine mammals (*b*), and most consistently for the Delphinidae (brown diamonds) and Balaenopteridae (pink squares) families. The marine mammal host tree was calculated from the mitogenomes of representative species. The hierarchical clustering dendrogram of marine mammal skin microbial communities is based on the average Bray–Curtis dissimilarity per host species.

Similar correlations between host phylogeny and associated microbiota have been observed elsewhere in nature. These relationships include host phylogeny and animal gut microbiotas [42], where co-speciation between particular bacteria and their hosts is hypothesized to be related to animal diet, with the microorganisms performing specific functions related to digestive processes. In addition to functional evolutionary pressures, vertical transmission of the gut microorganisms from parent to offspring can act to maintain host-specific relationships [43]. Skin microbial communities are less studied across the animal kingdom, although recent studies identified evidence for host-skin microbial phylosymbiosis in ungulates and coral reef fish [17,18], suggesting that evolutionary pressures may drive microbial relationships in these vertebrates.

We observed that intraspecific variation (within marine mammal species) in the skin microbiotas of marine mammals was less than interspecific and interfamilial (between species and between taxonomic families of marine mammals) relationships. Thus, at broad scales, the evolutionary relationships among hosts appear to structure the composition of the marine mammal skin microbiota, a pattern that may reflect both evolutionary and biochemical/metabolic pressures presented to both the host and microorganisms. As marine mammals diversified in association with the marine habitat, their skin played an important role in protection from exogenous conditions and pathogenic microorganisms in the marine environment [44], which may have partly driven the pattern we observed. A consistent pattern in the data was the differential composition of the microbiotas between the Balaenopteridae (fin and sei whales) compared to the Delphinidae (bottlenose dolphin, pantropical spotted dolphin, rough-toothed dolphin, short-finned pilot whale and melon-headed whale). These families diverged around 50 Ma [45], with the Balaenopteridae containing the largest marine mammals that feed using baleen and with specialized anatomical features for engulfing large quantitates of prey. By contrast, the Delphinidae include a diverse composition of toothed whales with streamlined bodies with a pronounced rostrum, among other features.

In prior studies of humpback whales [15] and killer whales (*Orcinus orca*) [46], host geographical location contributed to skin microbial community variation. Although marine mammal skin microbiotas may be impacted by the oceanic location of the host, in the present study host location is confounded with host identity due to a lack of representation of skin from the same species sampled in both Hawai'i and the US east coast (table 1). Physiochemical and nutrient conditions certainly govern the growth and distribution of marine bacteria, and temperature and nutrients, among other parameters, are indeed distinct between the warmer, oligotrophic Hawai'ian waters and the cooler and more eutrophic US east coast. Comparing skin microbiotas of species from different ocean environments, and in the context of physiochemical and nutrient conditions, will shed light on the

role of geography in structuring marine mammal skin microbiotas. A comparison of seawater near the sampled animals was not available in this study. Future studies will incorporate this type of sampling to provide additional insight into the contribution of seawater microorganisms to differences in marine mammal skin microbiotas.

The SIMPER results demonstrate that there is specificity in the MED sequence types contributing to the species and family-specific skin microbiotas. For example, in the baleen whales, two distinct marine mammal-specific groups of Moraxellaceae and Cardiobacteriaceae contributed to the host family similarity. In the Physeteridae, *Psychrobacter* and Flavobacteriaceae were significant contributors to the skin microbiotas. The Phocidae were also significantly structured by *Psychrobacter*, but a different MED than those associated with Physeteridae. Additionally, the Phocidae also displayed significant contributions from members of the *Pseudoalteromonas* and *Pseudomonas*. *Pseudomonas* (MED12781) was the only MED sequence to overlap between families and was shared between the Delphinidae and the Phocidae families. These animals were sampled by different laboratories and were collected from distinct oceanic environments; thus it is unlikely that this bacterium was an introduced contaminant. Further studies are necessary to understand if this *Pseudomonas* MED12781 is a cosmopolitan-type skin associate of diverse marine mammals and if these cells are performing similar or distinct functions.

The functions of the marine mammal skin microbial community are still largely unexplored. These cells may assist in breaking apart dead skin, possibly by degrading keratin. Bacteria could also be associated with the degradation of fouling cells such as diatoms [46]. Bacterial cells may be providing immune-related services to the host, such as the production of antimicrobial peptides. The microbes may interact in a less direct relationship with the host by occupying space on the skin and thereby reducing competition and colonization by pathogens [47]. Many of these bacteria are related to cells previously identified on other species of marine animals, suggesting possible similar roles across these taxa. *Psychrobacter* and other members of the Moraxellaceae are common aerobic bacteria associated with humpback whale skin [14,15] as well as the surfaces of diverse fish [48], and thus these bacteria appear to be widespread associates of cetaceans and harbour seals. Cardiobacteraceae is a gammaproteobacterial family frequently recovered from human and animal clinical cases [49], but sequences were previously recovered in an exhaled blow from humpback whales [50] and now apparently healthy baleen whale skin. Cardiobacteraceae requires further study in natural (non-clinical) settings.

The study presented here provides an indication that host phylogeny plays a role in contributing to the skin microbiotas of marine mammals. However, there are caveats associated with this study that limit the extent of this understanding. Due to the opportunistic nature of sampling, sample sizes per species were limited. Additional samples (approx. 50 per species) could provide more understanding about skin microbiota variability between animals, as well as relationships between microbial communities and animal age and sex. Cetacean skin was generally collected via biopsy from the same anatomical area across individuals, yet additional samples from different body areas could provide insight into anatomical-based variability. Skin samples were collected over different months and years, and consistency in time of sampling is an ideal revision for future investigations. Sample processing followed the Earth Microbiome Protocols, which did not include the sequencing of positive or negative controls. Future studies should include these comparisons to confirm results, which are especially important for low microbial biomass samples. Lastly, while there are 129 extant species of marine mammal, this study represents only nine species from four families, and is mostly focused on cetaceans. Future work is, therefore, needed to confirm our preliminary findings within a larger phylogenetic context.

# 5. Conclusion

Skin lesions and other conditions can reflect major health disorders in marine mammals [51,52]; hence, understanding the connection between the skin microbiota and regional and health-related factors could be important for health diagnoses, as well as management and conservation of marine mammals. Our finding that host taxonomy plays a role in structuring the skin microbiota of marine mammals can be used to guide and build knowledge about baseline skin microbial communities in marine mammals. This knowledge can facilitate future comparisons needed for investigations into the changing environmental and health-related conditions that may be influencing skin microorganisms and their hosts.

Ethics. Collection of skin samples was conducted under NOAA permits nos. 633-1483, 633-1778, 14097, 16325, 17670, 1071-1770-00 and 1000-1617.
Data accessibility. Sequences and metadata are accessible in the Earth Microbiome Project repository (https://qiita.ucsd.edu/emp/), study no. 1665 and EBI no. ERP016924 (see electronic supplementary material, table S1 for sample accessions).

Authors' contributions. A.A. designed the study; R.W.B., J.R., G.W., A.B. and S.L. contributed samples; L.W. processed samples; A.A., C.A.M., J.M.U., A.M.V.C. and M.S.L. analysed data; all authors contributed to the writing. All authors gave final approval for publication.

Competing interests. We declare we have no competing interests.

Funding. Funding provided by the Earth Microbiome Project, WHOI Marine Mammal Center, WHOI Ocean Life Institute and WHOI's Andrew W. Mellon Foundation Endowed Fund for Innovative Research to A.A. Hawai'i sampling was undertaken during field projects funded by grants from ONR (N000141310648 to R.W.B, N000141110612 to T.A. Mooney and N00014101686 to R.D. Andrews) and NMFS (NA13OAR4540212 to R.W.B).

Acknowledgements. We thank D. Rogers and D. Mattila, for facilitating sample collections, David Beaudoin for sample screening and the Earth Microbiome Project and especially Gail Ackerman for Illumina HiSeq sequencing. We thank several reviewers for their assistance in improving this manuscript.

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
