## [Reviewer comments · Royal Society Open Science]

Review History

RSOS-192046.R0 (Original submission)

Review form: Reviewer 1

Is the manuscript scientifically sound in its present form?

Yes

Are the interpretations and conclusions justified by the results?

Yes

Is the language acceptable?

Yes

Do you have any ethical concerns with this paper?

No

Have you any concerns about statistical analyses in this paper?

No

Recommendation?

Accept with minor revision (please list in comments)

Comments to the Author(s)

I am quite satisfied with the revisions. Thank you for taking everything into consideration so thoroughly. I only have one more request.

Line 126. Could you make clear whether your starting data set was 75 and after removal of samples with less than 3,000 reads you had less than 75 samples, or if the 75 is the number of samples that passed your QC? Could this info be included into Table 1 perhaps? It's minor, I know, but will help me to quickly get an idea of the sample sizes.

Review form: Reviewer 2 (Catherina Vendl)**Is the manuscript scientifically sound in its present form?**

Yes

Are the interpretations and conclusions justified by the results?

Yes

Is the language acceptable?

Yes

Do you have any ethical concerns with this paper?

No

Have you any concerns about statistical analyses in this paper?

Yes

Recommendation?

Accept with minor revision (please list in comments)

Comments to the Author(s)

The presented manuscript 'Marine mammal skin microbiomes are influenced by host phylogeny' contributes interesting new evidence to the field. It is well and clearly written and I appreciate the brevity of the manuscript. I would still like to suggest a few small changes.

It appears that the authors did not use seawater samples from the sites where they collected the skin samples of the marine mammals. It would have been interesting to compare the seawater microbiota to those on the skin. The authors detected an impact of location. Including seawater as controls would help to interpret these location-dependent differences. Could the authors comment on this? If they didn't collect seawater samples, it might be useful to add a sentence in the discussion that seawater samples should be included in the future.

In addition, it seems that the authors did not collect technical controls (blank DNA extraction, negative PCR product etc.). I am a big fan of including those in the analysis to rule out technical contamination. Although the authors mentioned that sample processing was performed in different labs, the use of the same DNA extraction kit could still have introduced consistent contaminants. Therefore, I would recommend including a comment that technical controls would be recommended in the future.

The authors appear to use the terms microbiota and microbiome interchangeably. According to a definition by Nature microbiota are characterised as follows: 'The microbiota is a collective term

for the micro-organisms that live in or on the human body. Specific clusters of microbiota are found on the skin or in the gastrointestinal tract, mouth, vagina and eyes' (<https://www.nature.com/subjects/microbiota>), whereas the microbiome is defined as ,The microbiome comprises all of the genetic material within a microbiota (the entire collection of microorganisms in a specific niche, such as the human gut). This can also be referred to as the metagenome of the microbiota' (<https://www.nature.com/subjects/microbiome>). As the authors only used the SSU rRNA gene for their analysis of the skin microbiota instead of the entire genome, I recommend to go with the term microbiota.

For bacteria, the family name should be italicized (e.g., line 253).

Decision letter (RSOS-192046.R0)

14-Apr-2020

Dear Dr Apprill

On behalf of the Editors, I am pleased to inform you that your Manuscript RSOS-192046 entitled "Marine mammal skin microbiomes are influenced by host phylogeny" has been accepted for publication in Royal Society Open Science subject to minor revision in accordance with the referee suggestions. Please find the referees' comments at the end of this email.

The reviewers and handling editors have recommended publication, but also suggest some minor revisions to your manuscript. Therefore, I invite you to respond to the comments and revise your manuscript.

- Ethics statement

- Data accessibility

If you wish to submit your supporting data or code to Dryad (<http://datadryad.org/>), or modify your current submission to dryad, please use the following link:
<http://datadryad.org/submit?journalID=RSOS&manu=RSOS-192046>

- Competing interests

- Authors' contributions

- Acknowledgements

- Funding statement

Because the schedule for publication is very tight, it is a condition of publication that you submit the revised version of your manuscript before 23-Apr-2020. Please note that the revision deadline will expire at 00.00am on this date. If you do not think you will be able to meet this date please let me know immediately.

- 1) A text file of the manuscript (tex, txt, rtf, docx or doc), references, tables (including captions) and figure captions. Do not upload a PDF as your "Main Document";

- 2) A separate electronic file of each figure (EPS or print-quality PDF preferred (either format should be produced directly from original creation package), or original software format);
- 3) Included a 100 word media summary of your paper when requested at submission. Please ensure you have entered correct contact details (email, institution and telephone) in your user account;
- 4) Included the raw data to support the claims made in your paper. You can either include your data as electronic supplementary material or upload to a repository and include the relevant doi within your manuscript. Make sure it is clear in your data accessibility statement how the data can be accessed;
- 5) All supplementary materials accompanying an accepted article will be treated as in their final form. Note that the Royal Society will neither edit nor typeset supplementary material and it will be hosted as provided. Please ensure that the supplementary material includes the paper details where possible (authors, article title, journal name).

If your manuscript is newly submitted and subsequently accepted for publication, you will be asked to pay the article processing charge, unless you request a waiver and this is approved by Royal Society Publishing. You can find out more about the charges at <https://royalsocietypublishing.org/rsos/charges>. Should you have any queries, please contact openscience@royalsociety.org.

on behalf of Dr Maximilian Telford (Associate Editor) and Kevin Padian (Subject Editor)
openscience@royalsociety.org

Associate Editor Comments to Author (Dr Maximilian Telford):

Associate Editor: 1

Comments to the Author:

The authors have rewritten this and made many changes suggested by the original authors. The new reviews (one by one of the original reviewers) suggest it can be accepted with some minor additional revisions. Most of these are trivial. One reviewer suggests a new statistical analysis (two alternatives are suggested) is required and the authors should undertake this.

Once these changes have been done then the paper would be acceptable for publication.

The comments regarding statistics were not in the 'comments to the authors' so I repeat them here.

"To my knowledge there are a number of issues with 'distance-based' analyses like PERMANOVA. They do not account for one of the main properties of multivariate data, the mean-variance relationship. One of the previous reviewers also commented on the use of PERMANOVA. Although the authors addressed this concern by adding Monte Carlo simulations within the PERMANOVA analysis, I would have liked the authors to either use a completely different analysis or at least to report on the results of PERMDISP, testing the differences in dispersion. If PERMDISP detected a significant difference across the individual samples, I am not sure if the addition of Monte Carlo simulations is sufficient to compensate for the difference in dispersion. Instead of a 'distance-based' analysis I recommend using the package mvabund in R. The mvabund approach improves power across a range of species with different variances and includes an assumption of a mean-variance relationship. It does this by fitting a single generalised linear model (GLM) to each response variable.

In addition, I would like the authors to explain their choice of MED as a method to cluster the bacterial sequences in opposition to using ASVs (Amplicon sequence variant), as it appears to me that ASVs are more commonly used in such analyses."

Associate Editor: 2

Comments to the Author:

Previous version was reviewed and rejected due to various problems with analyses and interpretation. Resubmission was encouraged but requiring a significant rewrite. It is an interesting data set that the reviewers thought would be of value if published.

Reviewer comments to Author:

Reviewer: 1

Comments to the Author(s)

I am quite satisfied with the revisions. Thank you for taking everything into consideration so thoroughly. I only have one more request.

Line 126. Could you make clear whether your starting data set was 75 and after removal of samples with less than 3,000 reads you had less than 75 samples, or if the 75 is the number of samples that passed your QC? Could this info be included into Table 1 perhaps? It's minor, I know, but will help me to quickly get an idea of the sample sizes.

Reviewer: 2

Comments to the Author(s)

The presented manuscript 'Marine mammal skin microbiomes are influenced by host phylogeny' contributes interesting new evidence to the field. It is well and clearly written and I appreciate the brevity of the manuscript. I would still like to suggest a few small changes.

It appears that the authors did not use seawater samples from the sites where they collected the skin samples of the marine mammals. It would have been interesting to compare the seawater microbiota to those on the skin. The authors detected an impact of location. Including seawater as controls would help to interpret these location-dependent differences. Could the authors comment on this? If they didn't collect seawater samples, it might be useful to add a sentence in the discussion that seawater samples should be included in the future.

In addition, it seems that the authors did not collect technical controls (blank DNA extraction, negative PCR product etc.). I am a big fan of including those in the analysis to rule out technical contamination. Although the authors mentioned that sample processing was performed in

different labs, the use of the same DNA extraction kit could still have introduced consistent contaminants. Therefore, I would recommend including a comment that technical controls would be recommended in the future.

The authors appear to use the terms microbiota and microbiome interchangeably. According to a definition by Nature microbiota are characterised as follows: 'The microbiota is a collective term for the micro-organisms that live in or on the human body. Specific clusters of microbiota are found on the skin or in the gastrointestinal tract, mouth, vagina and eyes' (<https://www.nature.com/subjects/microbiota>), whereas the microbiome is defined as 'The microbiome comprises all of the genetic material within a microbiota (the entire collection of microorganisms in a specific niche, such as the human gut). This can also be referred to as the metagenome of the microbiota' (<https://www.nature.com/subjects/microbiome>). As the authors only used the SSU rRNA gene for their analysis of the skin microbiota instead of the entire genome, I recommend to go with the term microbiota.

For bacteria, the family name should be italicized (e.g., line 253).

Author's Response to Decision Letter for (RSOS-192046.R0)

See Appendix A.

Decision letter (RSOS-192046.R1)

Dear Dr Apprill,

It is a pleasure to accept your manuscript entitled "Marine mammal skin microbiotas are influenced by host phylogeny" in its current form for publication in Royal Society Open Science. The comments of the reviewer(s) who reviewed your manuscript are included at the foot of this letter.

on behalf of Dr Maximilian Telford (Associate Editor) and Kevin Padian (Subject Editor)
openscience@royalsociety.org

Associate Editor Comments to Author (Dr Maximilian Telford):
Comments to the Author:

All remaining comments by reviewers have been dealt with satisfactorily. Recommend acceptance of the manuscript.

Appendix A

Response to reviewers

RSOS-192046

Marine mammal skin microbiomes are influenced by host phylogeny

Amy Apprill et al.

Comments from the authors are below each reviewer comment in **bold**.

Associate Editor Comments to Author (Dr Maximilian Telford):

Comments to the Author:

The authors have rewritten this and made many changes suggested by the original authors. The new reviews (one by one of the original reviewers) suggest it can be accepted with some minor additional revisions. Most of these are trivial. One reviewer suggests a new statistical analysis (two alternatives are suggested) is required and the authors should undertake this.

Once these changes have been done then the paper would be acceptable for publication.

Thank you for this positive response!

The comments regarding statistics were not in the 'comments to the authors' so I repeat them here.

"To my knowledge there are a number of issues with 'distance-based' analyses like PERMANOVA. They do not account for one of the main properties of multivariate data, the mean-variance relationship. One of the previous reviewers also commented on the use of PERMANOVA. Although the authors addressed this concern by adding Monte Carlo simulations within the PERMANOVA analysis, I would have liked the authors to either use a completely different analysis or at least to report on the results of PERMDISP, testing the differences in dispersion. If PERMDISP detected a significant difference across the individual samples, I am not sure if the addition of Monte Carlo simulations is sufficient to compensate for the difference in dispersion. Instead of a 'distance-based' analysis I recommend using the package mvabund in R. The mvabund approach improves power across a range of species with different variances and includes an assumption of a mean-variance relationship. It does this by fitting a single generalised linear model (GLM) to each response variable.

Thank you for this concern. We did conduct the PERMDISP comparisons, as suggested. According to PERMDISP, differences in microbiota dispersion in our comparisons at the family and species were not significantly different, thus providing stronger support for the significant results of our PERMANOVA analyses. We added these results in Table 1 as well as a new Table 3 in the main text. Additionally, pair-wise PERMDISP results comparing species were added to Supplementary Table 3. In Supplementary Table 3, the PERMDISP results were significant for two comparisons (Sei whale and Pantropical spotted dolphin, $p = 0.022$; Short-finned pilot whale and Pantropical spotted dolphin, $p = 0.048$). Thus, we adjusted our results to reflect the low confidence in the PERMANOVA results for these two comparisons.

We decided to remove the statistical tests on animal location entirely from the manuscript. This is because location is confounded in host identity. That is, there were no samples in the same family or species available from both geographic locations. Instead of presenting this result, we instead leave the topic for the Discussion: *'Although marine mammal skin microbiotas may be impacted by oceanic location of the host, in the present study host location is confounded with host identity due to a lack of representation of skin from the same species sampled in both Hawaii and the U.S. East Coast (Table 1). Physiochemical and nutrient conditions certainly govern the growth and distribution of marine bacteria, and temperature and nutrients, among other parameters, are indeed distinct between the warmer, oligotrophic Hawaiian waters and the cooler and more eutrophic U.S. east coast. Comparing skin microbiotas of species from different ocean environments, and in the context of physiochemical and nutrient conditions, will shed light on the role of geography in structuring marine mammal skin microbiotas. Comparison of seawater near the sampled animals was not available in this study. Future studies will incorporate this type of sampling to provide additional insight into the contribution of seawater microorganisms to differences in marine mammal skin microbiotas.'*

In addition, I would like the authors to explain their choice of MED as a method to cluster the bacterial sequences in opposition to using ASVs (Amplicon sequence variant), as it appears to me that ASVs are more commonly used in such analyses."

We recently spoke to Dr. Callahan, the developer of DADA2, the program that clusters into amplicon sequence variants (ASVs), about the differences between the clustering methods. Minimum entropy decomposition (MED) nodes are, in fact, amplicon sequence variants and provide relatively comparable results. DADA2 handles the sequencing errors differently than MED such that there would almost certainly be greater 'resolving power' with ASVs. A student in our lab compared the two methods on a 16S rRNA gene amplicon dataset and found similar trends in her data. We acknowledge that we may have missed rare variants and may have detected more false positives with MED than we would have with DADA2 (according to Callahan et al. 2016). We conducted these analyses prior to the release of DADA2 and as such, chose to use MED because it was the only method at the time that produced amplicon sequence variants. As such, we believe that MED produced solid results and that the potential benefits that could be gained by the greater resolving power of DADA2 would not necessarily outweigh the time and cost of re-doing the entire set of analyses.

Callahan, B. J., McMurdie, P. J., Rosen, M. J., Han, A. W., Johnson, A. J. A., Holmes, S. P. 2016 DADA2: high-resolution sample inference from Illumina amplicon data. *Nature methods*. 13, 581.

Comments to the Author:

Previous version was reviewed and rejected due to various problems with analyses and interpretation. Resubmission was encouraged but requiring a significant rewrite. It is an interesting data set that the reviewers thought would be of value if published.

Thank you for encouraging the rewrite. We believe the paper is much improved and appreciate the thoughtful suggestions.

Reviewer comments to Author:

Reviewer: 1

Comments to the Author(s)

I am quite satisfied with the revisions. Thank you for taking everything into consideration so thoroughly. I only have one more request.

Line 126. Could you make clear whether your starting data set was 75 and after removal of samples with less than 3,000 reads you had less than 75 samples, or if the 75 is the number of samples that passed your QC? Could this info be included into Table 1 perhaps? It's minor, I know, but will help me to quickly get an idea of the sample sizes.

Thank you for this question. Indeed, our final dataset had 75 samples. We clarified that in the text and it is also present in the title of Table 1.

Reviewer: 2

Comments to the Author(s)

The presented manuscript 'Marine mammal skin microbiomes are influenced by host phylogeny' contributes interesting new evidence to the field. It is well and clearly written and I appreciate the brevity of the manuscript. I would still like to suggest a few small changes.

Thank you for your positive comments.

It appears that the authors did not use seawater samples from the sites where they collected the skin samples of the marine mammals. It would have been interesting to compare the seawater microbiota to those on the skin. The authors detected an impact of location. Including seawater as controls would help to interpret these location-dependent differences. Could the authors comment on this? If they didn't collect seawater samples, it might be useful to add a sentence in the discussion that seawater samples should be included in the future.

We added this statement to this Discussion: '*Comparison of seawater near the sampled animals was not available in this study. Future studies will incorporate this type of sampling to provide additional insight into the contribution of seawater microorganisms to differences in marine mammal skin microbiotas.*'

In addition, it seems that the authors did not collect technical controls (blank DNA extraction, negative PCR product etc.). I am a big fan of including those in the analysis to rule out technical contamination. Although the authors mentioned that sample processing was performed in different labs, the use of the same DNA extraction kit could still have introduced consistent

contaminants. Therefore, I would recommend including a comment that technical controls would be recommended in the future.

Thank you for this comment. We added the following statement to the Discussion: *'Sample processing followed the Earth Microbiome Protocols, which did not include sequencing of positive or negative controls. Future studies should include these comparisons to confirm results, which are especially important for low microbial biomass samples.'*

The authors appear to use the terms microbiota and microbiome interchangeably. According to a definition by Nature microbiota are characterised as follows: 'The microbiota is a collective term for the micro-organisms that live in or on the human body. Specific clusters of microbiota are found on the skin or in the gastrointestinal tract, mouth, vagina and eyes' (<https://www.nature.com/subjects/microbiota>), whereas the microbiome is defined as 'The microbiome comprises all of the genetic material within a microbiota (the entire collection of microorganisms in a specific niche, such as the human gut). This can also be referred to as the metagenome of the microbiota' (<https://www.nature.com/subjects/microbiome>). As the authors only used the SSU rRNA gene for their analysis of the skin microbiota instead of the entire genome, I recommend to go with the term microbiota.

Throughout the manuscript, the term 'microbiota' has replaced our use of 'microbiome'.

For bacteria, the family name should be italicized (e.g., line 253).

All bacterial family names are now italicized.